# Facial Recognition System to Detect Student Emotions and Cheating in Distance Learning

**Fezile Ozdamli [1,2,\*], Aayat Aljarrah [2,3,\*], Damla Karagozlu [4] and Mustafa Ababneh [2,3]**

[1] Department of Management Information Systems, Near East University, Nicosia 99138, Cyprus
[2] Computer Information Systems Research and Technology Centre, Nicosia 99138, Cyprus
[3] Department of Computer Information Systems, Near East University, Nicosia 99138, Cyprus
[4] Department of Management Information Systems, Cyprus International University, Nicosia 99258, Cyprus
[\*] Correspondence: fezile.ozdamli@neu.edu.tr (F.O.); 20194007@std.neu.edu.tr (A.A.)

**Abstract:** Distance learning has spread nowadays on a large scale across the world, which has led to many challenges in education such as invigilation and learning coordination. These challenges have attracted the attention of many researchers aiming at providing high quality and credibility monitoring of students. Distance learning has offered an effective education alternative to traditional learning in higher education. The lecturers in universities face difficulties in understanding students' emotions and abnormal behaviors during educational sessions and e-exams. The purpose of this study is to use computer vision algorithms and deep learning algorithms to develop a new system that supports lecturers in monitoring and managing students during online learning sessions and e-exams. To achieve the proposed objective, the system employs software methods, computer vision algorithms, and deep learning algorithms. Semi-structural interviews were also used as feedback to enhance the system. The findings showed that the system achieved high accuracy for student identification in real time, student follow-up during the online session, and cheating detection. Future work can focus on developing additional tools to assist students with special needs and speech recognition to improve the follow-up facial recognition system's ability to detect cheating during e-exams in distance learning.

**Keywords:** cheating detection; computer vision algorithms; deep learning; distance learning; facial recognition

## 1. Introduction

Distance learning via online resources is a process that continues to support students and lecturers in teaching and learning around the world. Students can master successful self-directed learning skills with the aid of technology and educational resources online. They may determine what they need to learn, locate and use online resources, apply the knowledge to solve the classroom tasks, take exams, and even assess the feedback that comes as a result [1].

Students can acquire a degree through distance learning after successfully going through the exam process using online technology and resources, such as a learning management system (LMS) [2]. A facial recognition system is one of the technologies used for online learning to detect exam cheating and malpractices among students during distance learning [3]. A facial recognition system is often used to verify users through ID verification services. It operates by identifying and measuring the facial characteristics of a given student image or video. It compares a human face from an electronic picture or a video frame to a database of faces [3].

Students who use distance learning technology are more engaged than those who take a more traditional approach, and as a result, they spend more time on fundamental learning tasks [4]. Distance learning gives lecturers a chance to invigilate students so that

they reach their full potential and perform well in exams without getting involved in malpractice.

Although distance learning has numerous advantages, it still has many weaknesses, such as an inability to detect students' emotions, which play a huge role in student success [5]. In online lessons, lecturers cannot identify the emotions of their students or even the problems that they face because of the distance between them. They also still lack systems that enhance the credibility of online exams [6].

Due to the significance of this topic, many solutions and methods have been adopted to help lecturers accurately identify their students' emotions through algorithms that can detect emotions, such as face reader [7,8] and X press engine [9]. There have been many successful artificial intelligence algorithms, especially deep learning algorithms, which have proved popular in the computer vision field with the help of convolutional neural networks (CNN) [10]. It has been generally utilized in image recognition and classification. It must be noted that the process of face detection and recognition requires high accuracy in image processing considering its characteristics in making the system highly efficient and credible.

This research offers very significant methods in distance learning, which has recently increased in popularity. Despite various educational institutions using distance education as an alternative to traditional education, many have questioned the efficiency and effectiveness of the system [11].

The key problems identified by this study are the lack of a system to detect and verify students' identities in distance learning, online learning, and exams. This is in addition to the lack of a recognizable system to detect cheating attempts by students during online tests and quiz sessions. These issues have also been suggested in earlier research as significant obstacles to distance learning [12,13]. Online proctoring or invigilating has been a topic of much debate since its increased use during the COVID-19 pandemic for students' exams [3,14,15]. As part of their strategic planning, universities around the world are considering digital solutions that can be adopted, kept, or improved to check students' cheating behaviors on online exams.

To solve these problems, the objective of this study is to use computer vision algorithms and deep learning algorithms to develop a new system that supports lecturers in monitoring and managing students during online learning sessions and exams. The main focus is on the students' face identification, classroom coordination, facial emotion recognition, and cheating detection in online exams. The system detects, measures, and outputs the students' facial characteristics. The effectiveness of the system on students' facial emotion recognition during exams is assessed by lecturers through the interview. In addition, the system assists students with identity verification (facial authentication). It also included a cheating detection system that uses face detection, gaze tracking, and facial movement. The system also detects the appearance of more than one face in the image or focuses on the appearance of materials in use in the image, such as a paper, book, phone, or movement of the hand and fingers, which could be one of the mechanisms of cheating in the test. In addition, this system has a simple interface that provides easy access to each of these tools. This system was developed based on semi-structured interviews with lecturers to identify the obstacles and challenges they face during online or distance learning. Their views were also taken, and their comments were referred to on the system at all stages of system development until the system reached its latest design. Then, after completing the system in its final form, their opinion was taken to measure their satisfaction with the system and their need for such systems to be applied in distance learning strategies.

## 2. Related Studies

A detailed explanation of concepts and topics related to this study is provided in this section. This forms the key background for understanding this study, in addition to relevant methodological research, as shown in Table 1.

## 2.1. Face Identification

Face detection is considered a very early step that must be taken before the process of recognition can begin. The face detection technique was established in the 1970s. It has gradually improved alongside modern tools that can be used in real time [16].

Many systems have been developed to detect the target faces in places that contain surveillance cameras using the best software such as MATLAB and Python to process images in such systems. This software is very effective in fingerprint systems and many others, such as pattern matching analysis systems that are used for feature extraction, segmentation, and all detection and identification processes [17,18].

Facial recognition in real time and its percentage of accuracy are the basic aspects that should be improved. Facial recognition in online or distance learning is described as authenticating users to be real or fake individuals in attempting to become real users.

## 2.2. Sentimental Learning

### 2.2.1. Classroom Coordination

This is the method that lecturers use in the classroom to manage several activities in a multi-constraint context [19]. Lecturers have a very significant role in classrooms. They should consider recent modern educational models in their teaching. Currently, the classroom is considered an emotional atmosphere where pupils have both positive and negative emotions. Thus, it is very important to overcome the problems of passivity and stress by creating suitable emotions to facilitate learning. The classroom is an area where emotions and learning exist. Therefore, such a mix should provide secure, emotionally engaging environments that challenge and test as well as encourage knowledge acquisition [20].

### 2.2.2. The Importance of Emotions in Education

Emotions and sentimentality are very significant because they play an important role in the process of learning [21]. They help students acquire new kinds of knowledge and skills. In addition, they also have a positive role in allowing students to achieve academic commitment and success in school. Students who experience negative emotions such as being anxious or frustrated will be more disturbed in their exam performance than students who face positive emotions [22].

## 2.3. Face Emotion Recognition

Discovering emotions by relying on facial expressions is considered an act of human nature. It is the most intuitive way to detect human emotions on the face. The term "facial expression recognition" has been developed [23]. Sown used automatic analysis to detect facial expressions by following the motion spots in an image sequence. In that experiment, a correlation between all the expressed and experienced emotions was demonstrated [24]. Discovering modern techniques where machines become capable of understanding the expression of gestures and putting them into the context of feelings must be the key to communication with humans. Thus, several approaches to detecting emotions from the face were adopted. Such approaches tackle the main groups of emotions by relying on very simple static models that have successful results [25]. Other ways to successfully categorize information from facial expressions include relying on explicit code systems that resemble the dynamics in which facial movements are coded in a group of action units (AUs) with a muscular basis called the Facial Action Coding System (FACS) [26]. Many researchers have considered the dynamics of changing faces, since the automatic captures are covered by action units from facial motion sequences [27].

Face tracking has become a feature of many face-based emotion-recognition systems and many tools, including measurement of optical flow [28]. In addition to active contour models [29], face identification, recovery of a facial pose following facial expression [30], probabilistic approaches to detecting and tracking human faces [31], active [32] and adaptive [33] appearance models, multiple Eigenspaces-based methods [34], robust

appearance filters [35], and facial motion extraction based on optical flow [36], there are several other methods. Thus, the deep learning framework clarifies many classifiers used in several tasks of facial expression recognition. Each one has its advantages and disadvantages when dealing with the recognition problem, such as Support Vector Machines, Bayesian Network Classifiers, Linear Discriminant Analysis, Hidden Markov Models, and Neural Networks; for more detail, see [37–40]. Thus, some approaches can be highlighted in the facial expression of emotion, such as the parametric models extracting the shape of the mouth and its movements, in addition to the movement of both the eyes and eyebrows developed in [41], where the major directions of precise facial muscles are classified in the emotion recognition system treated in [42], and permanent and transient facial features such as lips, nasolabial furrows, and wrinkles, which are considered as recurrent indicators of emotions [43]. Finally, the last steps include the facial expression classification to estimate emotion-related activities.

### 2.4. Face Recognition in Online Learning

Facial recognition is used in many fields, such as human–computer interactions and artificial intelligence [44]. For example, facial recognition is used in online or distance learning with login systems and verification that the account is a real person and not a fake. As seen in Figure 1, the stages performed in the facial recognition process are described in most sections by relying on the use of 3WPCA-MD (Three-Level Wavelet Decomposition Principal Component Analysis with Mahalanobis Distance). This technique captures the extraction features of the PCAs, including three levels of the wavelet decomposition.

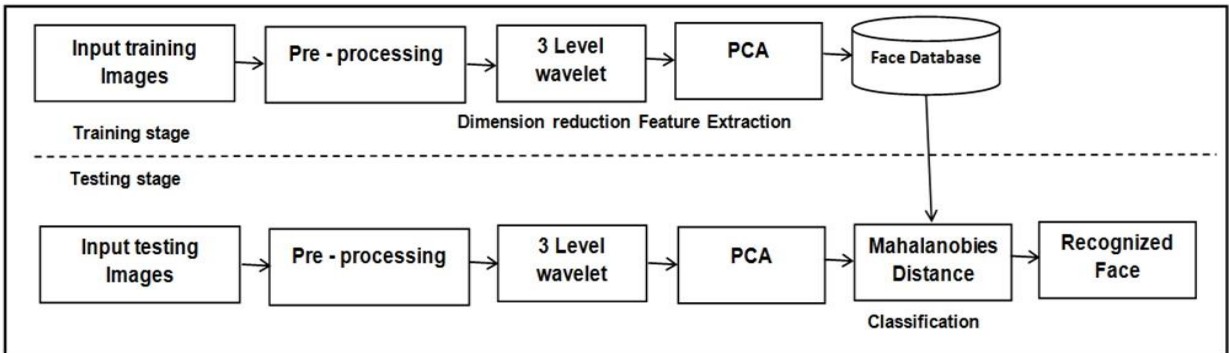

**Figure 1.** Preprocessing facial recognition.

However, the classification features use Mahalanobis Distance [45]. Thus, the results of such a tool were very good in an experiment by Winarno et al. [46], where the level of reliability could reach 95.7%, which led to faster results in the facial recognition calculation, especially when compared to the usual PCA methods. Thus, the average computational velocity value relying on the 3WPCA-MD tool is about five to seven milliseconds for every facial recognition process because of the support of Mahalanobis Distance Classification. This Mahalanobis Distance is a prominent approach used to develop the results of classification produced by exploiting the data structure [47]. Another study conducted by Winarno et al. [48] shows how the 3WPCA is used to expect fake face data with recognition accuracy that reaches more than 98% and provides a logical attendance system hinging on facial recognition.

### 2.5. Face Recognition System Architecture

Facial recognition systems analyze the data that are received or recorded from images or videos, which is undertaken in two steps, which are face extraction and detection [49]. The faces detection mechanism operates by identifying specific points in the face from the image. This process is called "feature extraction," through which different points

are identified from one face to another. Facial recognition systems are distinguished in their work even if the images of faces continue to flow or the face continues to move as well, by comparing the extracted features with features saved in the databases. Some examples of faces can be seen in [50].

### 2.6. Detecting Cheating in Online Exams

The students' performance in the courses is the basic criterion for higher education institutions to give them academic grades, and the evaluation of their performance in these courses depends on the grades they achieve in tests. Thus, students feel great pressure and a huge responsibility to pass these tests by whatever means [51,52]. As the credibility of these tests under such difficult circumstances worries the educational and administrative staff in the school [53,54], it is necessary to hold examinations by adhering to strict control standards to ensure credibility and efficiency [55,56].

Many models are used to classify abnormal activities in exams by using Multi-Class Markov Chain Latent Dirichlet Allocation. Such models are extremely helpful because they are used as feature detectors to discover the arm joints, shoulders, and head movement or direction. Thus, this model can classify the activities into five possible classes. It can also use a supervised dynamic and hierarchical Bayesian model for this [57]. Thus, many studies proposed methods to identify abnormal behaviors from the targeted videos, where the basic task is detecting abnormal activities. The proposed method helps detect any abnormal activity by discovering normal activities and classifying the rest as abnormal [58].

Many attempts have been made to detect cheating in online exams by relying on webcam analysis of an examinee's attitude. It works by checking the time and the student's head movement on a computer screen [59]. By using a kind of RGB camera, the analysis uses algorithms such as Support Vector Machine, depth sensors, and wearable tools popular in recognition. Thus, it is very apparent that the Kinect sensor or depth sensor in a human activity recognition system provide accurate findings [60].

In summary, Table 1 displays the review of studied models in the literature and the number of datasets used.

**Table 1.** Summary of previously published models and the number of datasets used.

| Paper | Aim | Model | Data Set | Result |
|---|---|---|---|---|
| [61] | This study aimed to build an automatic system to recognize facial expressions in real time. | k-Nearest Neighbor algorithm (k-NN). | 63,000 labeled images. | The results demonstrated the efficiency of emotion recognition using the k-NN algorithm in real-time. |
| [62] | This study aimed to develop an automatic system to recognize students' feelings during a learning session. | CNN model. | (FER2013) dataset. | The system achieved high accuracy in classifying feelings and identifying students' feelings. |
| [63] | This paper aimed to test some machine learning models to verify their efficiency in recognizing the human emotions. | Several machine learning models. | 13,584 images from the (HBCU) dataset and 16,711 from the (UC) dataset. | The results showed the effectiveness of using machine learning algorithms to monitor and identify human emotions with an accuracy similar to human observers. |
| [64] | In this paper, a system was developed to detect student behavior in online tests in real-time. | Decision Tree (DT), Naive Bayes (NB), and Support Vector Machines (SVMs). | 229 images from the (ROSE) datasets and 10,000 images from the (SMOTE) datasets. | The results showed that SVM has better accuracy than other algorithms to detect student behavior in online tests in real-time. |

| [65] | In this research, a system was developed for tracking student attendance in crowded classes using a smartphone camera. | Deep learning model based on the ResNet architecture. | 1750 images of about 70 students in 25 sessions. | The results demonstrated the effectiveness of using the facial recognition system to track student attendance and prevent fake attendance, which saves time and effort for the lecturers and students. |

## 3. Methodology

In this section, the general methodology used to develop this system with all its tasks is explained, in addition to adopting the Agile methodology [66]. Firstly, to begin to analyze the system requirements, the opinions of lecturers and experts were taken to identify the problems and challenges that they face in distance learning. Secondly, the contents were designed and drafted, and the experts' opinions were taken again about the system objectives and contents. Thirdly, development occurred in several stages, and the designers returned at each stage to take opinions and comments to improve the development and design of the system. Fourthly, after the design and development of the system was completed, the experts tested all system phases. Finally, the system deployment and feedback tracking phase involved taking the experts' opinions again about the system tasks. A semi-structured interview [67] was conducted with lecturers and experts to verify their satisfaction with and need for this system and to reveal the effectiveness and accuracy of this system. This section also discusses how to collect data, algorithms, and tools used to identify the student's face and feelings and then to detect cheating on an e-test. It is worth noting that the Agile methodology was adopted because it is characterized by its ease of change and easy modifications. It also saves time and effort by not requiring developers to completely rework any changes [68].

This system employs software methods, computer vision algorithms, and deep learning algorithms (i.e., convolutional neural networks) to develop a new system that gives lecturers the ability to properly coordinate a classroom and manage communication with their students during the lesson, in addition to ensuring their attention and studying their behavioral state in classes. Furthermore, many tools were presented to verify identity using facial feature extraction, as well as a system for detecting cheating by using face detection and gaze tracking. Moreover, this system has a simple interface that provides easy access to each of these tools. Figure 2 shows the system development process by an using the Agile method.

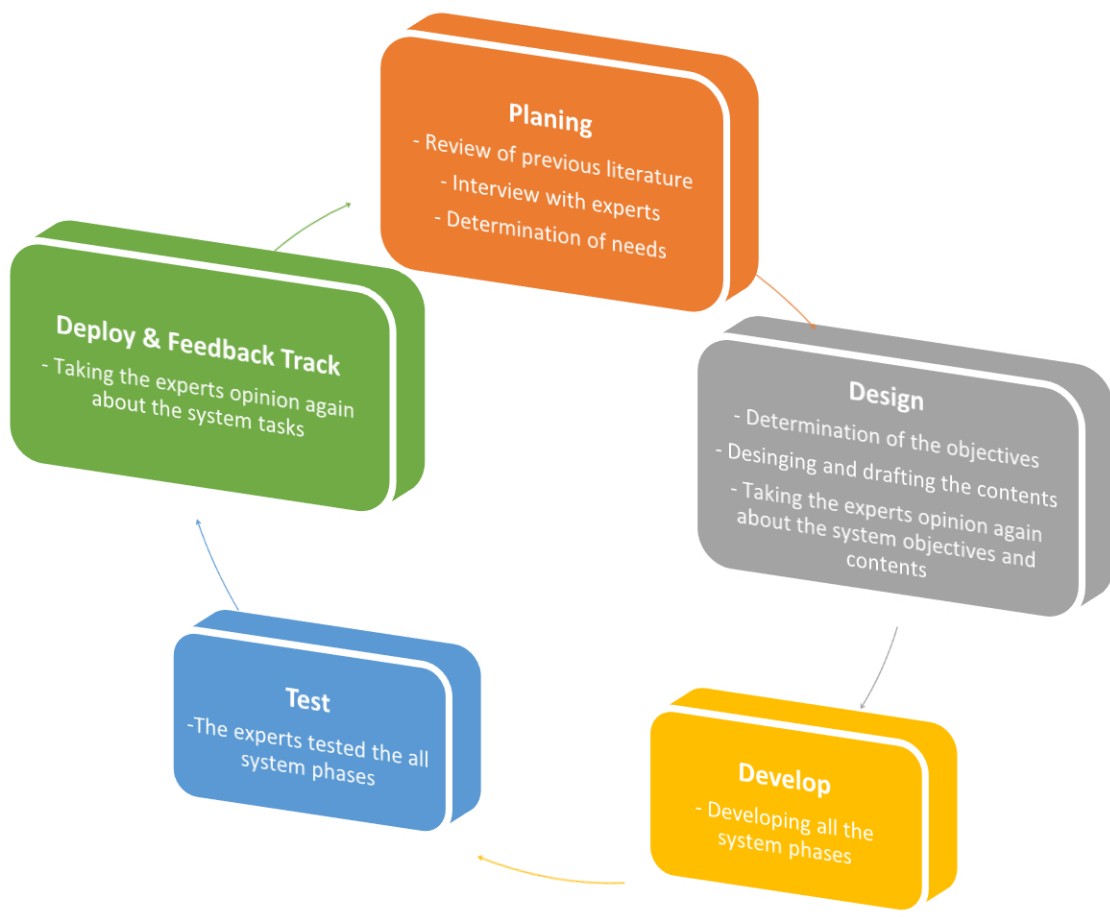

**Figure 2.** Agile development process.

### 3.1. Computer Vision Algorithms

A high-level language-written computer system was utilized to build and implement a computer vision algorithm. The OpenCV Library is a robust source of computer vision algorithms [69]. Presently coded in C++, OpenCV is free and open-source software. With such a wide variety of embedded vision processors (OpenCV), the algorithm analysis in this study focused on Agile to maximize pixel-level processing within the system. Within the Vision System Toolbox, Agile functions and Simulink blocks were utilized to implement computer vision algorithms, and they also allowed us to build our own library of functions that were tailored to the Agile programmable architecture.

The functions applied in computer vision system include:

- Image acquisition: Numerous image sensors from light-responsive cameras work together to create digital images. The image data produced were an image series. The pixel values represent the intensity of light in one or more spectral bands.
- Pre-processing: In order to ensure that the data meet the rules presented by the computer vision approach, the data had to be processed before the method could be applied to image data in order to extract the required information. To ensure that the image organization system was accurate, resampling was carried out. To ensure that sensor noise did not contribute to erroneous information, noise cutback was carried out. To ensure that important information could be detected, contrast augmentation was conducted. To improve visual formations at proximate relevant scales, a scale space representation was created.

- Feature extraction*: From the data, image characteristics of varying degrees of complexity were recovered. These consisted of ridges, lines, and edges. We then performed localized interest locations, such as blobs.
- Detection/Segmentation*: During the processing, a decision was made regarding which image points or areas should be processed further by dividing an image into segments, each of which included a different object of interest. Image partition into nested scene architecture was carried out that included the foreground, object groups, individual objects, or visual salience. Subsequently, we created a foreground match for each frame of one or more videos, while maintaining the continuity of the videos' temporal semantics.
- High-level processing: At this stage, the input often consists of a small amount of data, such as a collection of points or an area of an image that is supposed to contain a certain object. The data's compliance with model-based and application-based rules was checked in this section. The estimation of factors particular to a given object size was performed. Image recognition categorizes captured objects into many groups. Image registration compares and combines two images of the same object from different angles.
- Decision making*: The final decision is made by applying automated inspection programs via a Pass/Fail system, with a match or no-match in the recognition of images.

### 3.2. Convolutional Neural Network (CNN)

CNN as a deep learning system was used to take in an input image, rank various features and objects within the image, and distinguish between them. In this study, the [70] method of Mini Xception CNN analysis was utilized to statistically predict facial emotions. One section of the article describes how CNN is used to identify facial expressions of emotion. CNN analyzes real-time video frames to forecast the likelihood that each of the seven fundamental emotional states will occur. The analysis model uses real-time input from the CNN model's output data to predict future emotions in students. The two-phase hierarchical system makes it easier to examine human behavior and make predictions about the future. The seven fundamental facial emotions of the recognized face were statistically analyzed as the next phase after face detection. In the second and most important section of this study, which deals with emotion prediction, the derived data were used as input for the real-time model. A csv file containing the computed data was sent as the input. The model estimates the percentage of each emotion for the expected future using the data gathered about a student through time and a separate set of inputs for each emotion.

### 3.3. Data Collection and Tools

In this study, several datasets were used.

The first set of labeled faces in the raw data used to train face verification models was obtained from images available on the Internet, and this database contains approximately 13,500 images [7].

The second set is the dataset used to train emotion recognition models. These data were taken from the FER dataset database on Kaggle.com, which includes around 32,300 images of faces with different facial expressions. The data clarify seven major emotions at 48 × 48 pixels in grayscale [71].

The third dataset was used to train the gaze tracking model. This dataset was obtained from gaze interactions for everybody (GI4E) and consists of 1380 images with a resolution of 600 × 800 pixels [72]. The fourth dataset was used to train the head movement model. The dataset was obtained from the FEI Face Database and consists of 2800 images of 200 people at 640 × 480 pixels [73]. The fifth dataset was used to detect objects depicted in the camera. The dataset was obtained from common objects in context (COCO) and consists of 80 object categories [74].

### 3.4. Combining the Models to Develop the System

The [75] approach was used to merge the various models in order to analyze the datasets. The method represented the surfaces of images in the class using the potent statistical tools of 3D morphable models. Additionally, it merges two or more 3DMMs that are constructed from various datasets and distinct templates, some of which may only partially intersect. The tools combine two or more models by filling in the gaps in one model's data with data from the other using a regressor. The Gaussian Process framework is commonly used to combine covariance matrices from other models. This is done by creating a new face model that includes the variability and facial characteristics.

### 3.5. System Structure

This system was developed on a laptop running Windows 10 using Python 3.3.8. In addition, this system was developed by dividing it into three systems to achieve its desired goals and to meet the needs and suggestions of lecturers. Dividing it in this way makes it easier for us to develop and modify any system and reduces the processing effort for the computer. In this section, we present the architecture of systems, models used, and the algorithms. Figures 3 and 4 show the user interface and the system stages, respectively.

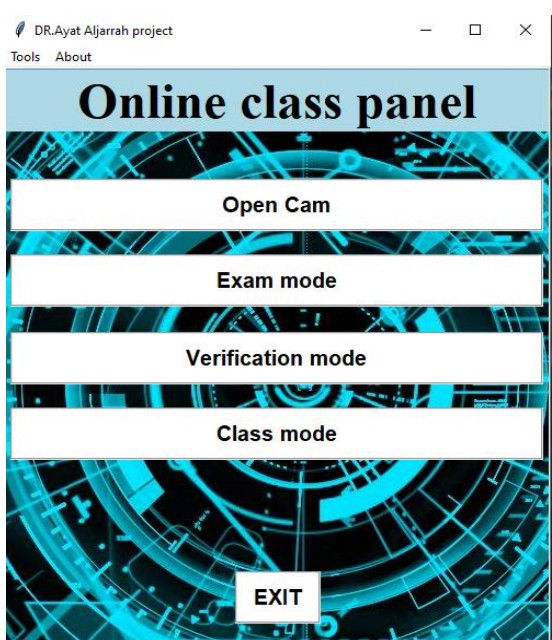

**Figure 3.** The user interface.

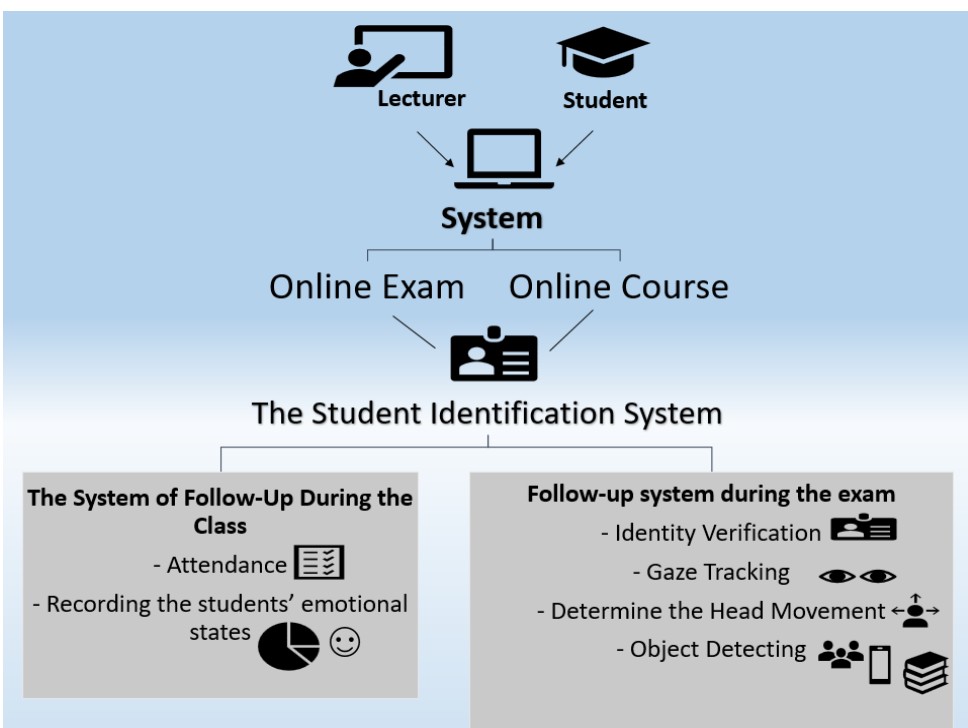

**Figure 4.** The developed system structure.

### 3.5.1. The Student Identification System

Referring to this system, students who attend the class or an examination will be identified remotely to verify whether the person who attends the lesson or the exam is the same as the account holder to avoid cases of cheating by identity theft.

Using such a system, the stored images of students must be compared to the images received via the web camera. As a result, it employs the deep metric learning algorithm and the ResNet-34 algorithm in two stages [76,77]:

1. The first stage: facial feature extraction from the images in the database.
2. The second stage is comparing the faces received via the webcam with the faces stored or existing in the database.

In the beginning, the pictures of the faces that must be recognized are entered, which are considered the photos of the class students. Therefore, the face is recognized, for instance, by detecting the presence or even the absence of faces and discovering their location. Then, about 128 facial features are extracted through an algorithm in the form of numerical coefficients and stored in a beam. "Deep metric learning" is an algorithm that depends on adopting an output represented by a beam of integer values instead of the usual tool with deep learning that accepts a single image as input and where the output is a classification or description of the picture [78].

Then, the resulting vector from 128 measures is compared with the vectors stored in the database, and if no match is made with any vector, the face is considered unknown, but if the vector is matched with an existing vector in the database, it will draw a square around the student's image, writing the student's name as it is saved in the database. The architecture of this facial recognition network depends on applying the 34-ResNet algorithm [79]. The network was trained on a (LFW) dataset that contains around 13,250 images belonging to 5794 people, split into 70% of the training data and 30% of the test data [7].

3.5.2. Interviews

Conducting interviews is one of the most popular methods for gathering data in qualitative research [46]. Semi-structured interviews were utilized in this study to examine more about the specifications for a facial recognition system that will detect exam cheating by students as well as to acquire more insight on how effective the system is for online proctoring and distance learning.

A pre-determined list of questions was used in the semi-structured interviews (Table 1), although the interviewers (lecturers) were also requested to clarify any points and go into further detail about some topics. The benefit of semi-structured interviews is that the researcher has complete control over the data collection process. The setting for interviews is voluntarily, seeking the interviewers' consent and informing them about the confidentiality and anonymity of their participation. There were two forms of interviews conducted in this study: one was for the lecturers, and the other for the students. Both interviews were conducted to measure the lecturers' and students' perceptions or feedback towards the effectiveness of the system for facial identification, classroom coordination, face emotion recognition, and detecting cheating in online exams.

This system was created after interviewing the lecturers (Table 2) on their perceptions of facial recognition system usage in their teaching and learning processes, particularly in exams in online or distance learning. These responses and the identified problems were used to develop the systems. The responses included the difficulties and barriers they encounter and the efficiency level of the system in recognizing and detecting the students' cheating behaviors and emotional expressions. At every level of the system's development, up until the system reached the final stage after a 14-week period, the lecturers' opinions were taken into consideration. Overall, six university lecturers at Cyprus International University were interviewed using semi-structured interviewing techniques, in addition to the twenty students who were chosen at random from a population of 100 undergraduate students at Cyprus International University, from the Department of Management Information Systems in Nicosia. A simple sampling technique was employed. Then, after the system was completed, the lecturers' feedback was obtained to determine how effective and satisfied they were with it and whether it should be included in distance learning or online learning sessions.

**Table 2.** Interview questions used.

| Interview Questions | |
|---|---|
| **Lecturers** | **Students** |
| Is facial recognition technology used in universities? | Is facial recognition technology used in universities? |
| What can you say about the facial recognition system? | Do you like the facial recognition system? |
| Are you using facial recognition technology in your teaching and learning process? | Does facial recognition system help your verification process? |
| Do you think the facial recognition system can help you effectively invigilate your students in online classes and exams? | Is the facial recognition system used in your online class and exam? |
| Does the use of facial recognition increase the risk of false detection of student cheating? What are your expectations of the facial recognition system in teaching and learning? | How easy is the face verification system? |
| Do you think facial recognition should be fast enough or moderate? How will the use of facial recognition affect students' privacy? | How fast is the face verification system? |

| | |
|---|---|
| Does the system accurately recognize students' faces in distance/online learning? | Is facial recognition accurate enough for exam use? |
| Does the detected student's face match the student's details? | Does facial recognition accurately register you? |
| What is the most common facial emotion exhibited by the students? | Does facial recognition affect your privacy? |
| Can the system differentiate similar images? | How is facial recognition different? |
| What is the common gaze you notice among the students? | What do you think about the efficiency of the system? |
| What is the most common type of attempt made by students? Eye movement or facial movement? | Are you satisfied with the facial recognition system? |
| How fast is the student's face detected using this system? | What is your level of satisfaction? High, moderate, or low? |
| What do you think about the efficiency of the system? | |
| Are you satisfied with the facial recognition system? | |
| Did the facial recognition perform as well as expected in detecting the students' cheating behavior during online exams? | |
| What is your level of satisfaction? High, moderate, or low? | |

### 3.5.3. The System Follow-Up during the Class

In such a system, the emotional state of the learners or students is observed throughout the lesson, and the significance of this system is that it gives the lecturer the ability to discover the state of the students in lessons. Therefore, the lecturer focuses attention on and distributes questions to the students to maintain the largest possible amount of student attendance for as long as possible during the lesson and after the lesson ends. Thus, the system gives the lecturer statistical feedback about the emotional states experienced by a student throughout the lesson, which shows the extent the lecturers monitor the students and the material in use.

The sentiment analysis system works through two main stages: recognizing the face and verifying its emotions. To achieve this result, the following two techniques are used:

1. Haar feature-based cascade classifiers (OpenCV cascade classifier):

It is defined as an algorithm that detects the frontal projection of the face in the incoming images, and it is a superior algorithm in the field of continuous face identification in real time [68]. This system used a Haar cascade classifier in each part needed to detect the eyes or faces.

2. Mini_Xception CNN Model:

Here, we trained a CNN, which receives the detected face with dimensions of 48 × 48 pixels as input and predicts the probability of each of seven feelings, listed below, as the output of the final layer [69].

Thus, our mission was to classify every face based on their feelings, encoding the following:

(0 = Angry, 1 = Disgust, 2 = Fear, 3 = Happy, 4 = Sad, 5 = Surprise, 6 = Neutral).

The database is separated into two files, the first of which is train.csv, a file with two columns, one for feelings and the other for pixels.

The photos are converted from the scale [0,1] field by dividing them by 255, then subtracting 0.5, and multiplying by 2. This scales to the field [−1,1], which is considered the best field to be the input of the neural network in this kind of computer vision problem.

In short, all these steps were taken for collecting and preprocessing data using the algorithms for determining the location of the face and training the convolutional neural network to identify the feelings that the students experience during the educational session. In addition, the architecture of the sentiment analysis system depends on applying the Mini_Xception CNN Model [69], which was trained on a FER dataset that contains around 35,600 images, split into 70% of the training data and 30% of the test data [60].

### 3.5.4. Follow-Up System during the Exam

One of the most prominent issues and limitations that face distance learning is e-exams, as institutions resort to various methods to implement exams and ensure that there are no cases of cheating.

Therefore, in this section, we developed a new system to follow up the learners during the exam by warning them if they look outside the permitted exam space on the computer screen or exhibit abnormal behavior.

The principle of work was based on four steps: the first was to verify the face to ensure that the person is present in the exam using a trained model based on the deep metric learning algorithm and ResNet-34 algorithm; the second step was to follow the eyes of the students and determine their coordinates in the image using a trained model based on gaze tracking. The model [80] was trained on a (Gi4E) dataset that contains around 1380 images, split into 70% of the training data and 30% of the test data [72]. Through this, we could know the focus area of the student and thus have the ability to warn the student if they are looking outside the computer screen, such as looking at a cheat sheet, a book, or any other aid. The third step was determining the head movement using a model with Haar cascade algorithms, which operate when the frontal face is absent [69]; it was trained on a (FEI) face dataset that contains around 2800 faces belonging to 100 people, split into 70% of the training data and 30% of the test data [73]. The fourth step was detecting the presence of any material other than the face, for example, the presence of a mobile phone, paper, book, or hand; the fifth step is detecting the presence of more than one face in the image. The YOLO (V3) model, which was trained on the COCO dataset, was used in the fourth and fifth steps to detect objects depicted in the camera (such as a human, book, watch, iPad, and any other device) in real time [69], where this dataset contains 5000 images split into 70% of the training data and 30% of the test data [75].

### 3.6. Data Analysis

Statistics were used to examine how various research findings can be handled and defined mathematically, as well as how various inferences can be made from the outcomes of the analysis. The datasets were examined using statistical analysis. We used the descriptive analytic method to analyze the responses from these interviews. We organized the questions and responses into themes. The results were represented as percentages.

### 3.7. Ethical Approval

An ethics application was submitted to the Cyprus International University, Nicosia ethics committee. The approval letter was issued on 4 December 2021 for this study before the study was conducted.

## 4. Results and Discussion

Step 1:

Based on expert suggestions and opinions during the development of the student identification system, which verifies a student's identity based on the algorithms used, it

was found that it is easy to compare the existing face in the database with the face that is taken by the camera. In cases where the face is not recognized or matched, the results appear as "unknown." On the other hand, when the face is recognized, the name of the face appears as it is saved in the database. Thus, the result of the performance and accuracy for the deep metric learning algorithm and the ResNet-34 algorithm in this system is about 99.38% (Figure 5).

Adjustments were made after the experts provided their feedback about the performance of the system, as the experts and lecturers suggested that the system register the unknown student in a state of absence directly within the system. The optimization was made, which improved the efficiency of ResNet-34 and reduced the time required.

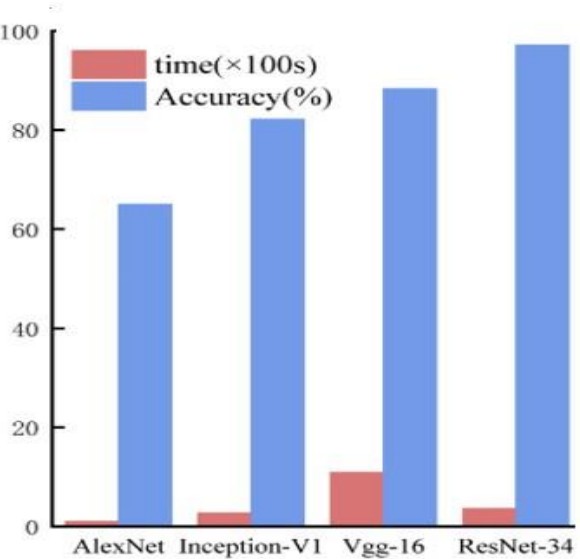

**Figure 5.** Performance and accuracy of deep metric learning algorithms.

A system to identify students and track their bibliometric attendance was also developed by [61] to support educational platforms; however, the accuracy of that system was only 95.38% compared to our system's 99.38%. In addition, our finding was closely related to the result previously reported by [81], in which the authors show testing accuracy of 88.92% with a performance time of fewer than 10 ms. This system was found to be suitable for identifying individuals in CCTV footage in locations because it masked facial recognition in real-time.

Step 2:

The success of distance learning systems depends on many factors, the most important of which is the extent to which lecturers understand the state of students' emotions and behaviors during the educational session, in order for the lecturers to adjust the teaching strategy in a timely manner to attract the attention of the largest number of students. Another factor is the ability of the lecturers to coordinate the students in classrooms during exams. Thus, based on expert suggestions and opinions during the development of the system of follow-up of the students in the class, it was developed to determine the feelings of the students depending on the deep learning algorithms (CNN). This system takes many consecutive photos for each student, shows the results graphically on screen, and also saves them. Therefore, the system gives the lecturer the ability to review the results after every session. It also allows the lecturer, in real time, to follow up with any student just by clicking on the student's name during the session.

When this system fails to find any faces, it notifies the student. This system is first in identifying faces. After six notifications, either the student's absence is noted for this session, or the decision is ultimately forwarded to the lecturer. Second, the system tracks each student's facial expressions and emotions throughout the session. At the conclusion of

each session, the system saves the remarks and feedback (from interviews) for each student in the form of a graph that displays the number of times they received messages, as well as other information. Additionally, the lecture is free login capability whenever they want to view student feedback. This system shows accuracy and high performance in discovering students' emotions, where after training, the model achieved an accuracy of 66% on average (Figure 6), despite the way of expressing the same emotion differing from one student to another. The authors in [62] developed a system to discover students' facial expressions and found that the accuracy of the system did not exceed 51.28%. The system achieved 62% accuracy in student follow-up during the online session [63]. Our results outperformed previous results by 66%, demonstrating improved student detection and verification in distance learning systems. Generally, this helps lecturers in the online sessions to know students' feelings effectively.

Moreover, changes made in response to the most recent expert advice suggest that the system should provide post-lesson feedback to all students on the same graph in order to serve as a session feedback chart according to the experts and lecturers. Experts and lecturers suggested renaming some facial expressions or emotions with other names related to the student's mental state during the lesson, for example, changing the name of the emotional state of sadness to "state of absent mind."

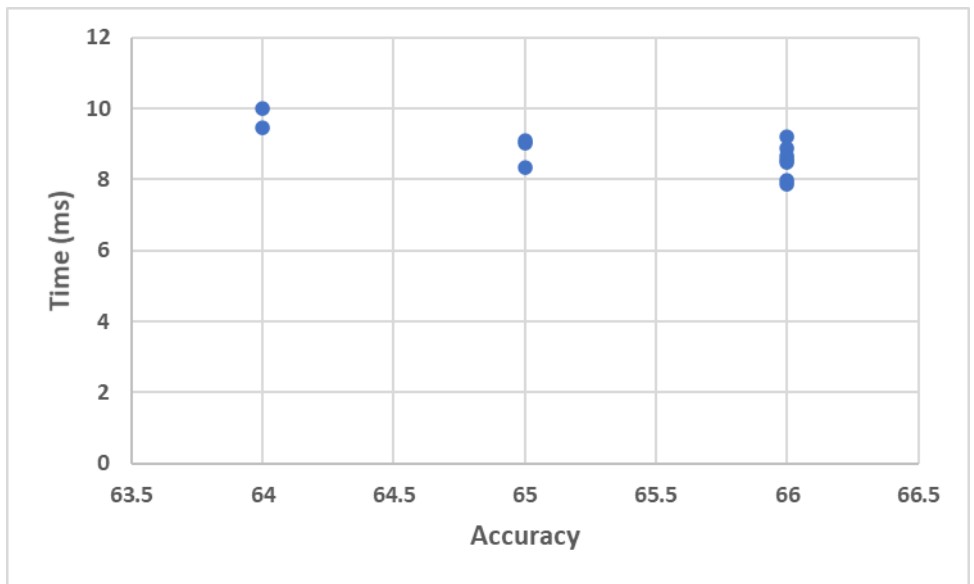

**Figure 6.** System accuracy against time.

Step 3:

The last section of this system addresses the most important factor in the success of distance learning systems, which is monitoring students' activities in e-exams. Thus, based on expert suggestions and opinions during the development of the system, the system was building a model to monitor students during the online exam by monitoring the iris of the eye based on a gaze tracking algorithm, and the complete absence of the face, as it monitors the movement of the entire eye. When the iris moves from the computer screen for more than five seconds, the system will send an audio alert to the student. If the alert is repeated more than five times, the student will be considered to be in a cheating situation. Similarly, if any of the following cases are observed, such as head movement, the presence of any material other than the face, or the presence of more than one face, the system will send an audio alert to the student.

Furthermore, the system makes a complete statistical profile for the situation of the student during that time. For example, if no alert is sent to the student during the exam, the framework will give a rundown after the end of the test that the student's condition was normal. Otherwise, when the student receives alerts, the system will provide

feedback showing the number of alerts and length of time of each abnormal case using the trained model YOLO (V3).

Figure 7 displays the results of performance and accuracy in terms of students' expressed behaviors. This system demonstrated the accuracy and high performance in detecting the abnormal behaviors of students in the e-exam, where the gaze tracking model achieved an accuracy of up to 96.95%, and the facial movement tracking model achieved up to 96.24%. In addition to that, the face detection and object detection models achieved an accuracy of around 60%. Moreover, happy behavior achieved an accuracy of 45.27%, while fear achieved an accuracy of 30.20%. From the lecturer's point of view, this system helps teachers in online exams to effectively identify expected cheating cases. Our findings were consistent with results of earlier research that studied mechanisms that students use to cheat in online exams [72,73], and their systems accurately (87.5%) tracked the movement of the student's head, face, and expressions of fear emotion during exams.

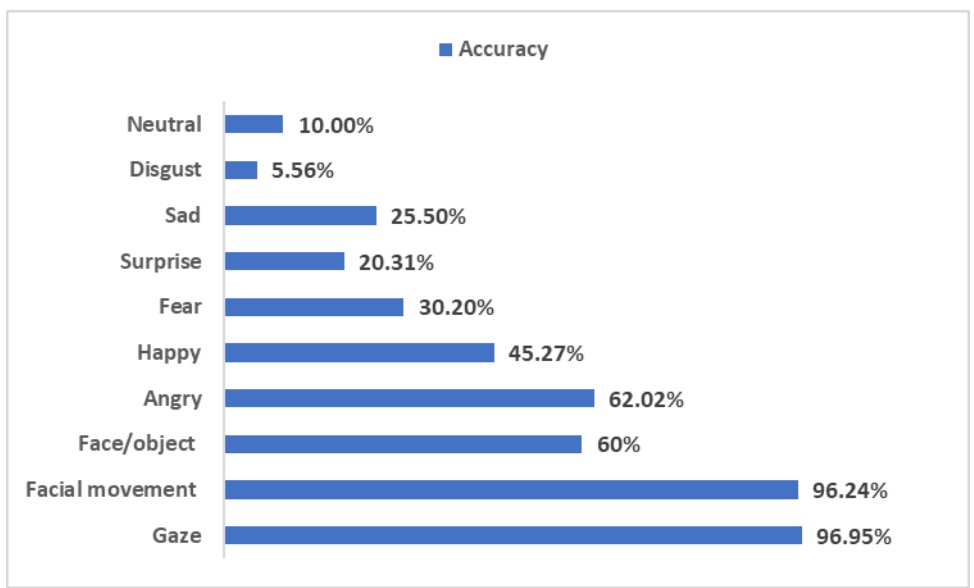

**Figure 7.** Performance and accuracy in terms of students' expressed behaviors.

Adjustments were made following the experts' feedback in which experts and lecturers suggested sending an alert to the exam invigilators in real time for each abnormal behavior (gaze, anger, facial movements, etc.) during e-exams, with a decision being made on the returned alert by the exam invigilators. Furthermore, the experts and lecturers suggested labelling each abnormal behavior descriptively in the feedback summary. At the end of the feedback, the system was finally developed based on the opinions and needs of lecturers for student invigilation. The opinions and comments of these reviewers (experts and lecturers) summarized the importance of having such integrated systems in distance learning in monitoring the students in e-exams. This is because the system was able to improve the educational process and increase the credibility of the e-exams, as well as save the lecturer time.

The quality of distance learning is ensured based on several factors, the most important of which is the lecturer's understanding of the needs of the educational tool or system that supports them in delivering high quality invigilation. The system will efficiently recognize students' faces and emotions and verify and detect their attempts in e-exams. The system developed here achieved this by improving the efficiency and credibility of e-testing systems. It also helps detect cheating in exams by recording students' feelings in real-time. This has been suggested as a solution for the issues with distance learning in research [70,71].

## 5. Conclusions

The findings of this study revealed that the facial recognition system achieved high performance and accuracy in detecting students' expressed behaviors, abnormal behaviors, gaze tracking, and facial movement tracking. The system was developed for invigilation purposes in distance learning. The system was aimed to be used by university lecturers to monitor students in e-exams, and it is equipped with a student verification system. Several deep learning algorithms were applied to develop this system, and the objective was achieved with high accuracy. However, during the development process, we faced challenges, such as real time data for cheating detection. Hence, we relied on lecturers' feedback in distance learning systems. The review of previous studies also provided us with information for the development of the system and analysis. Both forms of information were integrated to improve the system's efficiency and credibility for distance learning.

In conclusion, this study contributes knowledge of integrating different models to architecturally, imaginarily, and statistically analyze image data using deep learning algorithms and an agile approach to improve invigilation of students in distance learning systems. Although the effectiveness of distance learning and the credibility of e-exams could be improved with the help of this developed system, there were a number of drawbacks, such as the inability to automatically detect head and face movements; the inability to account for the various ways that a single student may express the same emotion to another; and the poor internet connection in rural areas. Future research could include new capabilities to help students with unique needs and speech detection to help the follow-up system during exams better identify cheaters.

**Author Contributions:** A.A. and M.A. designed and carried out the study and contributed to the analysis of the results and to the writing of the manuscript. F.O. and D.K. designed and carried out the study, collected data, and contributed to the writing of the manuscript. All authors have read and agreed to the published version of the manuscript.

**Funding:** The authors received no financial support for the research, authorship, and/or publication of this article

**Institutional Review Board Statement:** The study was approved by the Ethics Committee of Near East University.

**Informed Consent Statement:** Informed consent was obtained from all subjects involved in the study.

**Data Availability Statement:** The data presented in this study are available upon request from the corresponding author.

**Acknowledgments:** The participation of lecturers in this research is highly appreciated.

**Conflicts of Interest:** The authors declare that the research was conducted in the absence of any commercial or financial relationships that could be construed as a potential conflict of interest.

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
