# Peer review of "Facial Recognition System to Detect Student Emotions and Cheating in Distance Learning"

_sustainability, doi:10.3390/su142013230_

Round 1

Reviewer 1 Report

The manuscript describes the development and testing of a system that would support online learning by providing technical solutions to check student’s identities, their emotions and potential cheating behaviors during exams. While the system may be a valuable tool in the area of online learning platforms, there are several issues that need clarification in order to appreciate the value of the manuscript.

Critically, more methodological details are needed to evaluate the results of testing the proposed system. For instance, regarding the first step of student identification, the paper states (line 347) that “Thus, the result of the performance and accuracy for the deep metric learning algorithm and ResNet-34 348 algorithm in this system about 99.38%.” How was this result obtained? On how many participants or true/false exposures, in what type of experimental design? How was this verification study conducted? Was ethical approval for the study obtained?

The same comment is relevant for the other steps of model / system testing. For instance, the evaluation of the system’s accuracy in identifying student’s emotions would require the comparison of the predictions / assessments of the system with the reference point of students’ true experienced emotions. How was this performed in the study?

Several issues that need further clarification:

- line 59 “there are problems with identification” – what kind of problems?

- line 60 “how to establish an appropriate examination system” - what kind of problems?

- line 380 “The experts and teachers suggested to label some feeling in another names like label 380 the happy feeling to understand and label the sad feeling to absent-mind”

- line 50 the acronym CNN should be spelled out.

- line 203 at least a bibliographic reference in relation to the Agile methodology should be provided

There are also many misphrases in the manuscript:

- Abstract “to improve educational platforms and increasing” – should be increase

- Line 37 “The world will be witnessing will change”

- line 57 “After analyzed the previous literature”

- line 58 “there is some research argue”

- line 70 “which can be considered 70 wrong behavior indicates cheating incorrect behavior one of the mechanisms of cheating 71 in the test. ”

- line 76 “until the system reached the final images” – what images?

- line 92 - Face recognition in real time and its percentage of accuracy are from the basic aspects”

- line 247 “Where we analyzed the answers to these interviews using the descriptive research”

- line 374 “an average of 66% accuracy. and this 374 accuracy due to the differs way to express the same feeling from one student to another.”

There are many other phrases that need correction throughout the paper.

Author Response

Response to Reviewer 1 Comments

Point 1: For instance, regarding the first step of student identification, the paper states (line 347) that “Thus, the result of the performance and accuracy for the deep metric learning algorithm and ResNet-34 348 algorithm in this system about 99.38%.” How was this result obtained? On how many participants or true/false exposures, in what type of experimental design? How was this verification study conducted? Was ethical approval for the study obtained?

Point 1: How was this result obtained?

Response 1: The obtained results are taken from a published algorithm, and we used it in this research. We performed some experiments by the given information to get the results.

Point 2: On how many participants or true/false exposure?

Response 2: In each model different participants recorded are studied

Point 3: what type of experimental design?

Response 3: We simulate the algorithms, and we applied it on the used dataset. The experiments are evaluated according to the testing and training parts.

Point 4: How was this verification study conducted?

Response 4: We used standard evaluation measures to evaluate the proposed model such as accuracy of test data. Also, we compared the accuracy proposed method with other similar methods in the literature.

Point 5: Was ethical approval for the study obtained?

Response 5: No need for approval, as the data is available online for research purposes.

Point 6: The same comment is relevant for the other steps of model / system testing. For instance, the evaluation of the system’s accuracy in identifying student’s emotions would require the comparison of the predictions / assessments of the system with the reference point of students’ true experienced emotions. How was this performed in the study?

Response 6: We simulate the algorithms, and we applied it on the used dataset. The experiments are evaluated according to the testing and training parts.

Several issues that need further clarification:

- Point 7:  line 59 “there are problems with identification” – what kind of problems?

Response 7: Introduction Chapter improved by rewriting some paragraphs

- Point 8: line 60 “how to establish an appropriate examination system” - what kind of problems?

Response 8: Introduction Chapter improved by rewriting some paragraphs

Point 9: line 380 “The experts and teachers suggested to label some feeling in another names like label 380 the happy feeling to understand and label the sad feeling to absent-mind”

Response 9: The stated sentence (line 380) in Results and Discussion chapter has been rewritten.

Point 10: line 50 the acronym CNN should be spelled out.

Response 10: The stated acronym (line 50) in Introduction Chapter is clarified.

Point 11:  line 203 at least a bibliographic reference in relation to the Agile methodology should be provided

Response 11: A bibliographic reference has been added.

There are also many misphrases in the manuscript:

-Point 12: Abstract “to improve educational platforms and increasing” – should be increase

Response 12: Abstract improved by rewriting this sentence.

Point 13:  Line 37 “The world will be witnessing will change”

Response 13: The language of the paragraph corrected

Point 14:  line 57 “After analyzed the previous literature”

Response 14: The paragraph (line57) in Introduction chapter has been rewritten

Point 15: line 58 “there is some research argue”

Response 15: The paragraph (line 58) in Introduction chapter has been rewritten & clarified

Point 19: line 70 “which can be considered 70 wrong behavior indicates cheating incorrect behavior one of the mechanisms of cheating 71 in the test.”

Response 19: The paragraph (line 70) in Introduction chapter has been rewritten & clarified.

Point 20: line 76 “until the system reached the final images” – what images?

Response 20: The stated sentence (line 76) in Introduction Chapter is clarified

Point 21:  line 92 - Face recognition in real time and its percentage of accuracy are from the basic aspects”

Response 21: The paragraph (line 92) in Related Studies chapter has been rewritten

Point 22:  line 247 “Where we analyzed the answers to these interviews using the descriptive research”

Response 22: The paragraph (line57) in methodology chapter has been rewritten

Point 23:  line 374 “an average of 66% accuracy. and this accuracy due to the differs way to express the same feeling from one student to another.”

Response 23: The stated sentence (line 374) in Results and Discussion Chapter clarified

Point 24: There are many other phrases that need correction throughout the paper.

Response 24: The language of the article corrected by mdpi Language Editing Services.

Reviewer 2 Report

The authors seek to develop a new system to improve educational platforms and increase the quality of e-session learning in real-time. They claim that their system achieves high accuracy for student identification, student follow-up during the online session, and cheating detection. According to the authors, the system also recognizes the emotional states and motivation levels of students during an online session and monitors students during online exams. They further argue that this system can help to enhance distance learning platforms' efficiency and increase the credibility of e-tests, with some limitations that they will address in their future work. These limitations include not assessing for automatically face rotation and head movement, not addressing the different ways to express the same feeling from one person to another, etc.

I put my findings as under:

1.      The lack of a dedicated literature review section, the most important element for studies that are either based on or claim to explore and advance the current state of the matter, makes the article's organisation poor (as is the case of this study). The conclusion section should be condensed. The comparisons offered in the conclusion section might be moved to the section on the literature review. The comparison table that could display earlier research in this field, your work, and the importance of your work could be included in the literature review section.

2.      In the introduction section, please state the study objectives clearly.

3.      Short and with insufficient information, the Methodology section. The information regarding the teachers (as subjects, participants, or observers) is entirely absent. The essential information/plan regarding the procedures/methods, formulas, and tests used to carry out this study is not provided.

4.      The tests' application to the two datasets and the outcomes that were obtained may be covered in the Results section. How do authors get such high percentages as 66%, 99.38%, 96.95%, and 96.24%?

5.     The methodology section of the article would benefit greatly from being completely rewritten, as well as from having a dedicated literature review section as suggested in point 1 above. If the authors had included the study's goals and some additional potential future research directions, it would have been even better.

6.     This study's title is unattractive and somewhat misleading. Please give your model a fitting name and a meaningful title.

7.     The language is extremely awkward. Throughout the article's body, the authors reuse a significant number of sentences. The sentences themselves are very long. Unnecessary and meaningless words are used. I had a very difficult time understanding most of the sentences. The article needs to be completely rewritten and edited. As evidence, a certificate of editing might be offered.

Author Response

Response to Reviewer 2 Comments

Point 1: The lack of a dedicated literature review section, the most important element for studies that are either based on or claim to explore and advance the current state of the matter, makes the article's organization poor (as is the case of this study). The conclusion section should be condensed. The comparisons offered in the conclusion section might be moved to the section on the literature review. The comparison table that could display earlier research in this field, your work, and the importance of your work could be included in the literature review section.

Response 1: The Some of previous literature review table added.

Point 2: In the introduction section, please state the study objectives clearly.

Response 2: Introduction Chapter improved by rewriting some paragraphs and clarify the problem statements and objectives

Point 3: Short and with insufficient information, the Methodology section. The information regarding the teachers (as subjects, participants, or observers) is entirely absent. The essential information/plan regarding the procedures/methods, formulas, and tests used to carry out this study is not provided.

Response 3: Methodology section improved by rewriting some paragraphs

Point 4: The tests' application to the two datasets and the outcomes that were obtained may be covered in the Results section. How do authors get such high percentages as 66%, 99.38%, 96.95%, and 96.24%?

Response 4: Methodology section improved by rewriting some paragraphs and mentioned all datasets that used. Also, the obtained results are taken from a published algorithm, and we used it in our research.

Point 5: The methodology section of the article would benefit greatly from being completely rewritten, as well as from having a dedicated literature review section as suggested in point 1 above. If the authors had included the study's goals and some additional potential future research directions, it would have been even better.

Response 5: Study's goals and problem statement clarified in Introduction chapter, suggested table added in literature review and some paragraphs have been rewritten in Methodology section

Point 6: This study's title is unattractive and somewhat misleading. Please give your model a fitting name and a meaningful title.

Response 6: Title changed

Point 7: The language is extremely awkward. Throughout the article's body, the authors reuse a significant number of sentences. The sentences themselves are very long. Unnecessary and meaningless words are used. I had a very difficult time understanding most of the sentences. The article needs to be completely rewritten and edited. As evidence, a certificate of editing might be offered.

Response 7: The language of the article corrected by mdpi Language Editing Services.

Reviewer 3 Report

It is a rather descriptive study, rather than a research carried out with valid methods and tools.

Although the concept of real-time recognition of student behaviors is used in the title, the theoretical part presents approaches related to the recognition of emotions.

Research objectives and hypotheses must be clearly formulated.

Research requires a statistical analysis based on the use of advanced data processing methods.

Author Response

Response to Reviewer 3 Comments

Point 1: It is a rather descriptive study, rather than a research carried out with valid methods and tools.

Response 1: It is no the descriptive study it is development system study

Point 2: Although the concept of real-time recognition of student behaviors is used in the title, the theoretical part presents approaches related to the recognition of emotions.

Response 2: Title changed

Point 3: Research objectives and hypotheses must be clearly formulated.

Response 3: Introduction Chapter improved by rewriting some paragraphs and clarify the problem statements and objectives

Point 4: Research requires a statistical analysis based on the use of advanced data processing methods.

Response 4: There is no statistical data

Reviewer 4 Report

Generally, the topic is interesting, timely, and relevant. For the most part, the text is structured and well written. I have few suggestions throughout the text as below.

·        The introduction, para 1 is not relevant and needs to be re-written. The claim of technology and mathematics-related subjects should be backed up with some reference/s and also the covid thing has nothing to do with MOOCs. The authors need to clarify the focus as it is online (as in the title) or MOOCs. For example, an online course focuses more on content and MOOCs focus more on context. Again in para 2, distance learning came in. Please stick to the terminology and look at the differences between MOOCs and online degrees. There are many terms used so I suggest to be consistent with the terminology throughout the manuscript.

·        In lines (48 – 50), reference is not provided. 

·        The objectives are written in the problem statement (lines 60 – 64) which can be confusing for the readers.

·        Also the problem areas identified are too broad and even some of them (such as verification of student ID and attendance report) are well addressed with e-learning platforms such as Zoom and Microsoft Teams.

·    Online proctoring has been a topic of much debate since its increased use during the COVID-19 pandemic. As universities consider which digital solutions to let go, keep, or develop further in strategic planning, there are opportunities to evaluate usage and consider the growing body of evidence around the ethics, impact and effectiveness of tools like this. Please look at the following papers.

Coghlan, S., Miller, T. & Paterson, J. (2021). Good Proctor or “Big Brother”? Ethics of Online Exam Supervision Technologies. Philosophy and Technology, 34(4), 1581–1606. https://doi.org/10.1007/s13347-021-00476-1

Lee, K., & Fanguy, M. (2022). Online exam proctoring technologies: Educational innovation or deterioration? British Journal of Educational Technology, 53(3), 475–490. https://doi.org/10.1111/bjet.13182

Selwyn, N., O’Neill, C. Smith, G., Andrejevic, M., & Xin, G. (2021). A Necessary Evil? The Rise of Online Exam Proctoring in Australian Universities. Media International Australia. https://doi.org/10.1177/1329878X211005862

·        The connection of using various datasets and the development of the system is not clear at all. The systems development was informed by teachers’ interviews so when and how the datasets were used. My biggest concern is that the datasets cannot be generalised to the teachers/students for that institution so how they are connected.

·        How were the collected data pre-processed? Or it was already processed?

·        The application and language used to train the datasets is not mentioned.

·        The accuracies of the models addressed in the results section, don’t have any further explanation on how and which methods were used to get the accuracy outcomes.

·        The semi structured collected data in line 254, is a bit unclear how the interview data was collected and what types of questions.

·        The results and discussion section is weak as the steps are not cleared. It doesn’t seems like so called problem areas from section 1 are all addressed. I suggest this section to be re-written.

I wish the authors the best of luck as they move forward.

Author Response

RESPONSE TO REVIEWER COMMENTS

Reviewer 4

Response

Generally, the topic is interesting, timely, and relevant. For the most part, the text is structured and well written. I have few suggestions throughout the text as below.

Thank you

The introduction, para 1 is not relevant and needs to be re-written. The claim of technology and mathematics-related subjects should be backed up with some reference/s and also the covid thing has nothing to do with MOOCs. The authors need to clarify the focus as it is online (as in the title) or MOOCs. For example, an online course focuses more on content and MOOCs focus more on context. Again in para 2, distance learning came in. Please stick to the terminology and look at the differences between MOOCs and online degrees. There are many terms used so I suggest to be consistent with the terminology throughout the manuscript.

This section has been completely revised.

In lines (48 – 50), reference is not provided.

References have been added.

he objectives are written in the problem statement (lines 60 – 64) which can be confusing for the readers.

The objective has been revised accordingly.

Also the problem areas identified are too broad and even some of them (such as verification of student ID and attendance report) are well addressed with e-learning platforms such as Zoom and Microsoft Teams.

The problem statement has been revised.

Online proctoring has been a topic of much debate since its increased use during the COVID-19 pandemic. As universities consider which digital solutions to let go, keep, or develop further in strategic planning, there are opportunities to evaluate usage and consider the growing body of evidence around the ethics, impact and effectiveness of tools like this. Please look at the following papers.

Coghlan, S., Miller, T. & Paterson, J. (2021). Good Proctor or “Big Brother”? Ethics of Online Exam Supervision Technologies. Philosophy and Technology, 34(4), 1581–1606. https://doi.org/10.1007/s13347-021-00476-1

Lee, K., & Fanguy, M. (2022). Online exam proctoring technologies: Educational innovation or deterioration? British Journal of Educational Technology, 53(3), 475–490. https://doi.org/10.1111/bjet.13182

Selwyn, N., O’Neill, C. Smith, G., Andrejevic, M., & Xin, G. (2021). A Necessary Evil? The Rise of Online Exam Proctoring in Australian Universities. Media International Australia. https://doi.org/10.1177/1329878X211005862

Thank you for the suggestions. It has been revised. These references were helpful and used.

The connection of using various datasets and the development of the system is not clear at all. The systems development was informed by teachers’ interviews so when and how the datasets were used. My biggest concern is that the datasets cannot be generalised to the teachers/students for that institution so how they are connected.

The connection of using various datasets and the development of the system has been clarified and clearly stated in the revised version.

How were the collected data pre-processed? Or it was already processed?

The data were collected and processed before using. It has been stated in the revised file.

The application and language used to train the datasets is not mentioned

It has been revised accordingly.

The accuracies of the models addressed in the results section, don’t have any further explanation on how and which methods were used to get the accuracy outcomes.

The results have been further clarified and more explanations have been provided.

The semi structured collected data in line 254, is a bit unclear how the interview data was collected and what types of questions.

The interviews have been explained clearly in the revised version.

The results and discussion section is weak as the steps are not cleared. It doesn’t seems like so called problem areas from section 1 are all addressed. I suggest this section to be re-written.

The results and discussion section have been revised accordingly.

I wish the authors the best of luck as they move forward.

Thank you sir.

Round 2

Reviewer 1 Report

Some of my comments were addressed, but major revisions are still required.

The methodological information is still unclear and incomplete. Firstly, more details are needed about the images data sets on which the models were trained and then tested – which were the categories in which images were classified (in the original database and/or by the researchers performing this study)? On which criteria? How many images were in each category? Did the image sets include the same person in more than one image (which would account for between-subjects variability when checking for eye movements or emotion, for instance)? If yes, in how many images?

Secondly, more details are needed about the results of the models testing. Only an overall percentual indicator of performance in mentioned for step 2, although the model is supposed to identify six distinct emotions, plus the neutral state. How accurate was the model in identifying each emotion? This could be a valuable information for those interested in using the model in their practice. Moreover, it’s necessary to provide more information on the sensitivity and specificity of the models (in the framework of the Receiver Operating Characteristics approach), not only an overall percentage. More specifically, how often was the model able to correctly identify the target event (e.g., identify fear in an image where the protagonist truly expresses fear in step 2 or looking outside the computer screen in step 3), and how often was it able to identify the absence of the event (e.g., the neutral state in step 2 and looking at the screen in step 3)? Of course, any other metrics of model accuracy, more complex than these, are welcome, for instance those reported in the paper referenced at number 55 in the manuscript.

Another unclear issue: Line 376 states that “this system takes many consecutive photos for each student” – but previously the manuscript stated that the system / model was tested on pre-existing images, not on students.

There are still many misphrases, including in the newly added content (for instance line 354 “Where this dataset contains 5000 images split into 70% of the training data and 30% of the test data.”). Similarly, the correction made to the phrase in the abstract that I had remarked as misphrased is still wrong (i.e., “The purpose of this study is to develop a new system that should be increase”). I recommend an extensive language check.

Author Response

RESPONSE TO REVIEWER COMMENTS

Reviewer 1

Response

The manuscript describes the development and testing of a system that would support online learning by providing technical solutions to check student’s identities, their emotions and potential cheating behaviors during exams. While the system may be a valuable tool in the area of online learning platforms, there are several issues that need clarification in order to appreciate the value of the manuscript.

All the issues indicated have been resolved accordingly as suggested in the revised manuscript.

Critically, more methodological details are needed to evaluate the results of testing the proposed system. For instance, regarding the first step of student identification, the paper states (line 347) that “Thus, the result of the performance and accuracy for the deep metric learning algorithm and ResNet-34 348 algorithm in this system about 99.38%.” How was this result obtained? On how many participants or true/false exposures, in what type of experimental design? How was this verification study conducted? Was ethical approval for the study obtained?

The methodologies have been details as suggested.

Number of participants have been indicated.

The ethical approval information has been included in the revised version.

The same comment is relevant for the other steps of model / system testing. For instance, the evaluation of the system’s accuracy in identifying student’s emotions would require the comparison of the predictions / assessments of the system with the reference point of students’ true experienced emotions. How was this performed in the study?

The evaluation information has been provided.

Several issues that need further clarification:

- line 59 “there are problems with identification” – what kind of problems?

The statement has been revised accordingly.

- line 60 “how to establish an appropriate examination system” - what kind of problems?

The statement has been revised accordingly.

- line 380 “The experts and teachers suggested to label some feeling in another names like label 380 the happy feeling to understand and label the sad feeling to absent-mind”

The statement has been revised accordingly.

- line 50 the acronym CNN should be spelled out.

It has been spelled out as suggested.

- line 203 at least a bibliographic reference in relation to the Agile methodology should be provided

The reference has been provided.

There are also many misphrases in the manuscript:

- Abstract “to improve educational platforms and increasing” – should be increase

The statement has been revised accordingly.

- Line 37 “The world will be witnessing will change”

The statement has been revised accordingly.

- line 57 “After analyzed the previous literature”

The statement has been revised accordingly.

- line 58 “there is some research argue”

The statement has been revised accordingly.

- line 70 “which can be considered 70 wrong behavior indicates cheating incorrect behavior one of the mechanisms of cheating 71 in the test. ”

The statement has been revised accordingly.

- line 76 “until the system reached the final images” – what images?

The statement has been revised and information on images has been provided.

- line 92 - Face recognition in real time and its percentage of accuracy are from the basic aspects”

The statement has been revised accordingly.

- line 247 “Where we analyzed the answers to these interviews using the descriptive research”

The statement has been revised accordingly.

- line 374 “an average of 66% accuracy. and this 374 accuracy due to the differs way to express the same feeling from one student to another.”

There are many other phrases that need correction throughout the paper.

The statement has been revised accordingly.

The methodological information is still unclear and incomplete. Firstly, more details are needed about the images data sets on which the models were trained and then tested – which were the categories in which images were classified (in the original database and/or by the researchers performing this study)? On which criteria? How many images were in each category? Did the image sets include the same person in more than one image (which would account for between-subjects variability when checking for eye movements or emotion, for instance)? If yes, in how many images?

The information on methodology has been detailed.

Details on the images have been provided.

Secondly, more details are needed about the results of the models testing. Only an overall percentual indicator of performance in mentioned for step 2, although the model is supposed to identify six distinct emotions, plus the neutral state. How accurate was the model in identifying each emotion? This could be a valuable information for those interested in using the model in their practice. Moreover, it’s necessary to provide more information on the sensitivity and specificity of the models (in the framework of the Receiver Operating Characteristics approach), not only an overall percentage. More specifically, how often was the model able to correctly identify the target event (e.g., identify fear in an image where the protagonist truly expresses fear in step 2 or looking outside the computer screen in step 3), and how often was it able to identify the absence of the event (e.g., the neutral state in step 2 and looking at the screen in step 3)? Of course, any other metrics of model accuracy, more complex than these, are welcome, for instance those reported in the paper referenced at number 55 in the manuscript.

The results have been revised accordingly.

Another unclear issue: Line 376 states that “this system takes many consecutive photos for each student” – but previously the manuscript stated that the system / model was tested on pre-existing images, not on students.

The statement has been revised accordingly.

There are still many misphrases, including in the newly added content (for instance line 354 “Where this dataset contains 5000 images split into 70% of the training data and 30% of the test data.”). Similarly, the correction made to the phrase in the abstract that I had remarked as misphrased is still wrong (i.e., “The purpose of this study is to develop a new system that should be increase”). I recommend an extensive language check.

The statement has been revised accordingly.

The manuscript has been proofreading and corrected as suggested.

Reviewer 2 Report

I do not know why, but our comments were not taken seriously. Compared to the original, this revised version contains no significant changes. The term "Distance Learning" was changed to "Online Learning" in the title only, despite its frequent occurrence in the body of the paper (which was overlooked). It is important to note that if you simply reused the algorithms of other researchers, what is your contribution?

Author Response

RESPONSE TO REVIEWER COMMENTS

Reviewer 2

Response

The authors seek to develop a new system to improve educational platforms and increase the quality of e-session learning in real-time. They claim that their system achieves high accuracy for student identification, student follow-up during the online session, and cheating detection. According to the authors, the system also recognizes the emotional states and motivation levels of students during an online session and monitors students during online exams. They further argue that this system can help to enhance distance learning platforms' efficiency and increase the credibility of e-tests, with some limitations that they will address in their future work. These limitations include not assessing for automatically face rotation and head movement, not addressing the different ways to express the same feeling from one person to another, etc.

Yes, thank you

The lack of a dedicated literature review section, the most important element for studies that are either based on or claim to explore and advance the current state of the matter, makes the article's organisation poor (as is the case of this study). The conclusion section should be condensed. The comparisons offered in the conclusion section might be moved to the section on the literature review. The comparison table that could display earlier research in this field, your work, and the importance of your work could be included in the literature review section.

The literature review section has been added.

The conclusion has ben revised.

In the introduction section, please state the study objectives clearly.

The study objective has been clearly stated as suggested.

Short and with insufficient information, the Methodology section. The information regarding the teachers (as subjects, participants, or observers) is entirely absent. The essential information/plan regarding the procedures/methods, formulas, and tests used to carry out this study is not provided.

The methodology section has been revised accordingly.

The tests' application to the two datasets and the outcomes that were obtained may be covered in the Results section. How do authors get such high percentages as 66%, 99.38%, 96.95%, and 96.24%?

It has been revised accordingly.

The methodology section of the article would benefit greatly from being completely rewritten, as well as from having a dedicated literature review section as suggested in point 1 above. If the authors had included the study's goals and some additional potential future research directions, it would have been even better.

The methodology and literature review sections have been revised as requested.

Suggestions for the future studies has been provided accordingly.

This study's title is unattractive and somewhat misleading. Please give your model a fitting name and a meaningful title.

The title has been revised accordingly.

The language is extremely awkward. Throughout the article's body, the authors reuse a significant number of sentences. The sentences themselves are very long. Unnecessary and meaningless words are used. I had a very difficult time understanding most of the sentences. The article needs to be completely rewritten and edited. As evidence, a certificate of editing might be offered.

The revised manuscript has been proofread and edited. Thank you

Reviewer 3 Report

All improvements are qualitative. The table with the analysis of studies in the field is very well developed.

Author Response

RESPONSE TO REVIEWER COMMENTS

Reviewer 3

Response

It is a rather descriptive study, rather than a research carried out with valid methods and tools.

The methodology and tool used has been added in the revised version.

Although the concept of real-time recognition of student behaviors is used in the title, the theoretical part presents approaches related to the recognition of emotions.

The title has been modified and the content of the manuscript has been revised.

Research objectives and hypotheses must be clearly formulated.

The research objectives and hypotheses have been clearly stated in the revised version.

Research requires a statistical analysis based on the use of advanced data processing methods.

The statistical analysis used have been provided.

All improvements are qualitative. The table with the analysis of studies in the field is very well developed.

Yes, it is a qualitative study. Thank you

Reviewer 4 Report

Thank you for the revisions. The manuscript is significantly improved now. I wish authors best of luck :)

Round 3

Reviewer 1 Report

My previous comments were addressed.

Reviewer 2 Report

Your revision of the article and responses to my questions are satisfactory.

Thanks